# ComMU: Dataset for Combinatorial Music Generation

**Lee Hyun**[*][†]
Pozalabs
hyun@pozalabs.com

**Taehyun Kim**[*]
Pozalabs, Yonsei Univ.
taehyun@pozalabs.com
kimth0101@yonsei.ac.kr

**Hyolim Kang**
Yonsei Univ.
hyolimkang@yonsei.ac.kr

**Minjoo Ki**
Yonsei Univ.
minjoo@yonsei.ac.kr

**Hyeonchan Hwang**
Pozalabs
hyeonchan@pozalabs.com

**Kwanho Park**
Pozalabs
kwanho@pozalabs.com

**Sharang Han**
Pozalabs
sharang@pozalabs.com

**Seon Joo Kim**
Pozalabs, Yonsei Univ.
seonjoo@pozalabs.com
seonjookim@yonsei.ac.kr

## Abstract

Commercial adoption of automatic music composition requires the capability of generating diverse and high-quality music suitable for the desired context (e.g., music for romantic movies, action games, restaurants, etc.). In this paper, we introduce *combinatorial music generation*, a new task to create varying background music based on given conditions. *Combinatorial music generation* creates short samples of music with rich musical metadata, and combines them to produce a complete music. In addition, we introduce *ComMU*, the first symbolic music dataset consisting of short music samples and their corresponding 12 musical metadata for *combinatorial music generation*. Notable properties of ComMU are that (1) dataset is manually constructed by professional composers with an objective guideline that induces regularity, and (2) it has 12 musical metadata that embraces composers' intentions. Our results show that we can generate diverse high-quality music only with metadata, and that our unique metadata such as track-role and extended chord quality improves the capacity of the automatic composition. We highly recommend watching our video before reading the paper (https://pozalabs.github.io/ComMU/).

## 1 Introduction

Although musical composition is a creative process, algorithmic approaches for automatic music composition have been continuously studied [1, 2]. Recently, deep learning has shown great potential in composition. Bretan et al. [3] and Jaques et al. [4] have introduced deep sequence models into generating music sequences. After the seminal works, prior works [5–8] have proposed conditional music generation with initial sequences and musical metadata using language models. Such deep models have improved the quality of the composition and can create authentic music.

While the models continue to improve and generate authentic music, creating music on a commercially usable level still remains as an issue. Popular music is mostly a *homophony* in the sense that one track

---

[*]Equal Contribution
[†]Work done at Pozalabs, now at POSTECH EE

36th Conference on Neural Information Processing Systems (NeurIPS 2022) Track on Datasets and Benchmarks.

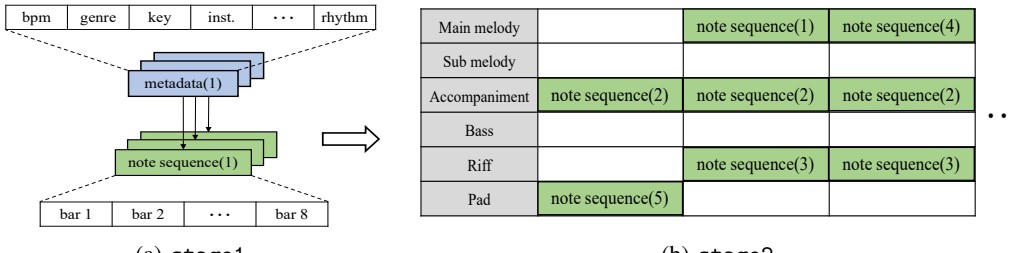

| | | | | | |
|---|---|---|---|---|---|
| Main melody | | | note sequence(1) | note sequence(4) | |
| Sub melody | | | | | |
| Accompaniment | note sequence(2) | note sequence(2) | note sequence(2) | | ··· |
| Bass | | | | | |
| Riff | | | note sequence(3) | note sequence(3) | |
| Pad | note sequence(5) | | | | |

(a) `stage1`  (b) `stage2`

Figure 1: **The whole process of *Combinatorial Music Generation***. In `stage1`, a note sequence ( green box) is generated from a set (blue box) of metadata. `Stage2` then combines note sequences generated from several metadata sets to create one completed music. The number of bars for a note sequence and the number of note sequences for a complete piece of music can be flexible (*ComMU* mostly has a note sequence of 4, 8, and 16 bars). In this work, we focus on solving `stage1`.

is in charge of the main melody while remaining tracks harmonically support the main track [9]. As the main melody and accompanying tracks are strictly separated, the common practice of composing the homophony music is *combinatorial*. In other words, tracks including the main melody and its accompaniments are separately generated and combined afterwards. Despite this lazy combination, the dissonance between them is prevented by the chord-conditioned generation of each track. In this context, we propose *combinatorial music generation*, which mimics the human composition convention on homophony music. Figure 1 shows the overall process of generating combinatorial music – track-level note sequences are created with a set of metadata, which are combined later to create a complete piece of music.

To generate commercially usable music, it is essential to have a detailed control over the generating process. For example, when a movie director requests fast and tense music suitable for an action scene, composers tend to pre-set proper musical metadata that affects the mood of the music — chord progression, key, genre, and rhythm — then compose music based on the metadata. From this point of view, we design `stage1` (Figure 1(a)) that generates note sequences, the elements for combinatorial music generation, with rich musical metadata to embrace composer's intention and harmony.

All in all, combinatorial music generation must be harmonized by vertically stacking note sequences within the homophony scheme and capture the intention of composition. Therefore, generating note sequences under elaborate musical controls such as track-role, instrument, and chord progression is necessary. However, previous symbolic music datasets [10–15] do not have enough metadata for sophisticated control. To tackle this issue, we present *ComMU*, a symbolic music dataset containing 11,144 MIDI samples that consist of short note sequences that are manually composed by professional composers with its corresponding 12 metadata (bpm, genre, key, instrument, track-role, time signature, pitch range, number of measures, chord progression, min/max velocity, and rhythm).

Our rich metadata can embrace desired musical conditions required for combinatorial music generation. It nicely controls the generated music by the given metadata and creates diverse music close to human creativity, leveraging numerous combinations of metadata at `stage1` (Figure 1(a)) and note sequences at `stage2` (Figure 1(b)). While previous tasks on music generation have centered on music continuation [16], reconstruction [17], style transfer [18], or creating music with a few metadata [19–24], we focus on composing diverse music with abundant metadata, similar to the way a composer generates music.

In this paper, we focus on `stage1`, the combinatorial music generation with the ComMU dataset. We evaluate the controllability, fidelity, and the diversity of generated note sequences. Our results show that (1) combination of multiple metadata can generate diverse and high-quality music with an auto-regressive language model, (2) the unique metadata (e.g., extended chord quality, track-role) improves the capacity and flexibility of the automatic composition.

Overall, the main contributions of this paper are:

- We propose the combinatorial music generation task with the ComMU dataset for the industry-level automatic music composition. Diverse and high-quality music is created with our framework and dataset.

Table 1: Comparison of ComMU to recent MIDI datasets with various metadata. We compare ComMU to other MIDI dataset on 4 types of metadata: genre, instrument, track-role, and chord progression.

| Dataset | Genre | Instrument | Track-role | Chord progression |
|---|---|---|---|---|
| Lakh MIDI [10] | ✓ | ✓ | - | - |
| MAESTRO [12] | ✓ | (✓)[1] | - | - |
| MSMD [13] | - | (✓)[1] | - | ✓ |
| ADL Piano MIDI [14] | ✓ | (✓)[1] | - | - |
| GiantMIDI-Piano [15] | - | (✓)[1] | - | ✓ |
| EMOPIA [23] | - | (✓)[1] | - | ✓ |
| **ComMU**(Ours) | ✓ | ✓ | ✓ | ✓ |

[1] include only one instrument.

- ComMU is the first symbolic music dataset manually created by professional composers for automatic music composition with 12 metadata.

- We show that our unique metadata such as track-role and extended chord quality are essential musical metadata for human-like composition, as they play a crucial role in expressing the comprehensive intention of the composer.

## 2 Related work

**Conditional music generation.** `Stage1` of the combinatorial music generation task can be considered as an extension of conditional music generation in that it generates a track with a given set of metadata. Many preceding works share similar scheme; for instance, MuseNet [6] creates music sequences in compliance with the given music style or instruments, and FIGARO [17] generates music with a few metadata that are extracted from a reference music. There are models [25–27] that can generate note sequences based on a given chord progression. They can produce music that fits the chord, but do not convey other important metadata such as rhythm and instrument.

MMM [19] is the closest task to ours, which takes instruments, bpm, and the number of bars as conditions and produces multiple instrument-tracks. However, MMM differs from our task in that the generated track cannot be combined into the homophony scheme with coherence, because it cannot take the track-role and the chord progression as conditions. It is rather specialized towards resampling or inpainting with a given music.

**Symbolic music dataset.** Various music datasets with metadata have been introduced with the development of conditional music generation. Although there are various forms of expressing music datasets, such as MIDI, audio, and piano-roll, we focus on MIDI-based datasets with metadata for the comparison with ComMU. Lakh MIDI Dataset [10], MAESTRO [12], and ADL Piano MIDI [14] have various meta-information such as genre, instrument, key, and time signature, but there is no chord information for the harmony of track-level composition. On the other hand, MSMD [13], GiantMIDI-Piano [15], and EMOPIA [23] have chord information that can infuse harmony, but lack metadata such as genre, and rhythm that can reflect the intention of composition.

Aside from the fact that existing symbolic music datasets do not have rich metadata, there are no datasets with track-role information. Although some datasets [10, 28, 29] define track as instruments, ComMU is the first dataset to separate the instrument from the track-role, allowing combinatorial music generation. We list major differences between ComMU with other datasets in Table 1.

## 3 ComMU dataset

### 3.1 Dataset collection

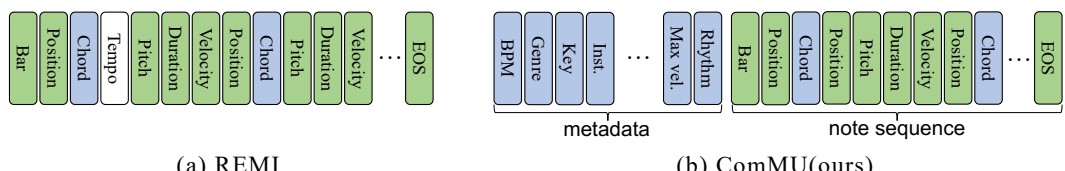

(a) REMI  (b) ComMU(ours)

Figure 2: Illustration of REMI and ComMU representation. ComMU differs from REMI in that it has metadata, extended chords and increased note resolution, and eliminates tempo token.

ComMU has 11,144 MIDI samples that consist of short note sequences with their corresponding 12 metadata. Basic information of ComMU dataset is given in Table 2.

Fourteen professional composers have manually made MIDI samples with metadata for 6 months. In detail, the composers were split into two teams. The first team created composition guidelines for certain genres and moods by finding and analyzing reference music. Each guideline then becomes an instruction (i.e., metadata) for composing samples, such as instruments, track-role, chord progressions, and pitch range available for each sample. The second team composed MIDI samples in accordance with each guideline, taking a total of 6 months to produce 11,144 music samples. The music made with these objective guidelines acquires regularity, suitable for the machine learning system.

Table 2: Basic information of ComMU.

| | |
|---|---|
| # SAMPLES | 11,144 |
| # NOTES | 526,612 |
| TYPES OF AUDIO KEYS | 24 |
| TYPES OF INSTRUMENTS | 37 |
| TYPES OF GENRES | 2 |
| TYPES OF RHYTHM | 2 |
| TYPES OF PITCH RANGE | 7 |
| TYPES OF TRACK-ROLE | 6 |
| TYPES OF TIME SIGNATURE | 3 |
| TYPES OF NUM-MEASURES | 3 |
| RANGE OF BPM | 35-160[1] |
| RANGE OF MIN VELOCITY | 2-127[2] |
| RANGE OF MAX VELOCITY | 2-127[2] |

[1] The range of representation is 5-200.
[2] Velocity 1 is for keyswitch note which is a key pressing when changing the playing type for each instrument.

## 3.2 Pre-processing and representation

When creating the dataset, deciding on the appropriate data representation is one of the most important considerations. Our representation is based on REMI [30], which encodes music samples in a token-based fashion. Throughout 12 metadata, REMI supports representing chord progression among 60 patterns, but we extend them to 108 patterns to hold diverse chord quality. Other 11 metadata are placed before the REMI representation. We also remove the tempo token as the tempo change never happens in our samples due to their short length. In addition, we increase the resolution of the position and the duration token from 32 notes in REMI to 128 notes to improve the quality of the result. See Appendix D.2 for an empirical study of the representation resolution. We compare REMI with our revised representation in Figure 2.

Unlike other metadata, the chord progression cannot be encoded as a single token due to its sequential form. Therefore, we encode the chord progression into a note sequence with position tokens. See Appendix A for more details about pre-processing and data representation.

## 3.3 Metadata

We provide the definition of 3 metadata which can be relatively ambiguous. Others including bpm, key, instrument, time signature, pitch range, number of measure, min/max velocity, and rhythm are explained in Appendix B.

**Genre.** We chose the new age and the cinematic genres for our dataset, which are often used in background music. We define the new age as a melodious genre mainly consisting of keyboard instruments and small-scale instruments such as acoustic instruments. The cinematic genre is a large-scale genre with orchestra, especially involving classical instruments such as string ensembles in charge of the melody and the accompaniment.

**Track role.** It is a classification of what role the created note sequences have in a multi-track music. We divide multi-track into main melody, sub melody, accompaniment, bass, pad, and riff.

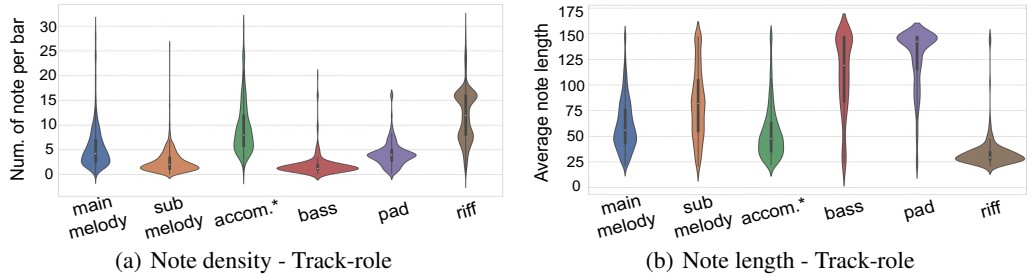

(a) Note density - Track-role    (b) Note length - Track-role

Figure 3: Illustration of the distribution in note density and note length, according to the track-role. Depending on the track-role, the shape of the corresponding notes varies. *accom.: accompaniment.*

**Chord progression.** Chord progression is the set of chords that are used in a sample. Extended chord quality leads to the improvement in harmonic performance and diversity by letting the model learn various possible melodies associated with the same chord progression.

### 3.4 Data analysis

In order to create music that is well-controlled only by the metadata, it is beneficial to have a distinguishable distribution of note sequences according to the metadata in ComMU. We extract MIDI-based features and analyze the distribution according to the metadata to observe the correlation between the metadata and the note sequences. In addition, we analyze the correlation as a heatmap, showing that a transformer [31] that covers the long-term dependency is suitable as a baseline. Among various options in our analysis, we present relevant features below.

**Analysis between metadata and notes.** We measure the characteristics of the note sequence by note density and length [23]. Note density is the average of the number of notes appearing in one bar, and note length is the average of the lengths of all notes in a sample. Figure 3 shows the distribution of note density and length according to the track category. Melody and accompaniment have short note lengths, whereas bass and pad have relatively long notes. Note density also shows a well-distinguished distribution of melody/accompaniment, bass/pad, and riff groups. This means that the musical dynamics of the melody/accompaniment are relatively strong, while the bass/pad have weak and stable notes. The riff track has a high density as the accompaniment track or higher. Considering it with the length, we can infer that the sound pattern with repeated short notes is the characteristic of the riff track.

**Analysis between metadata.** Figure 4 illustrates the correlation between instruments, track-role, and pitch range. Figure 4(a) shows that keyboards are frequently employed for the main melody and the accompaniment tracks, but pluck strings such as guitars are utilized more for the accompaniment and the riff tracks than for the main melody. In the case of lead instruments, it is rarely used in certain track roles. It can be seen that there is a significant correlation between track-role and instrument. Therefore, when music is generated under the condition of a given instrument without the track role information, the generated sequence is highly likely to be correlated with the track-role. This makes it difficult to make music with a low correlation track-role and musical instrument combination. For example, music with a keyboard as a bass and a guitar as a main melody is rarely produced without explicit track-role condition. We conduct an experiment to show this in Section 5.2.

Through the correlation between the track-role and the pitch range, Figure 4(b) demonstrates that the pitch range primarily used varies depending on the track-role. The pitch of the melody track is relatively high, while the accompaniment and pad tracks are distributed one level lower. The bass track has the lowest range of notes, following the common sense, and the riff track has a high pitch range similar to the sub melody. This is because the sub melody and riff tracks have the same role in supporting the main melody and mainly use the pitch range that the main instrument does not use. See Appendix C for more details about data analysis and limitations.

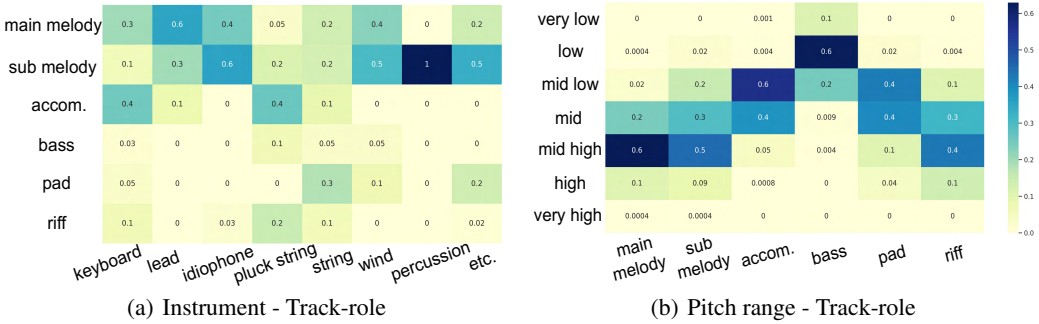

(a) Instrument - Track-role           (b) Pitch range - Track-role

Figure 4: Correlation heatmap of ComMU metadata. The instrument type and pitch range are different depending on the track-role.

## 4 Experiments

In this section, we examine *combinatorial music generation* with the ComMU dataset. For evaluation, we set 90% of the data for training and hold out the rest for validation. All the metadata in the validation set is exclusive to the training set. We apply the data augmentation method defined in Appendix A.1 to the training set.

### 4.1 Problem definition

Our task aims to generate music with specific musical metadata. Given a ComMU sample (Appendix, Figure 1) $X = \{x_1^M, .., x_{11}^M, x_{12}^S, .., x_N^S\}$, where $M$ and $S$ indicate the tokens ($x_n$) for the metadata and the note sequence respectively, and $x_N^S$ is the eos token, we train an auto-regressive language model by maximizing the following log-likelihood function.

$$\mathcal{L}_\theta(X) = \sum_{t=12}^{T} \log p_\theta(x_t^S \mid x_{<t}). \tag{1}$$

When the training is completed, we can generate diverse note sequences with a specific metadata $x_{1:11}^M$ and the chord progression $\mathcal{C} = \{x_i^{chord}\}_{i=1}^{K} \in x^S$ as follows:

$$\widehat{x_t^S} = g(p_\theta(x_t^S \mid x_{1:11}^M, \mathcal{C})), \tag{2}$$

where K is the length of the chord progression, and $g$ is a decoding algorithm. We use Top-k [32, 33] sampling for the decoding.

In music generation, the auto-regressive language model with the transformer structure has shown to be very effective [5–8]. Among them, we use Transformer-XL [34] as the baseline to illustrate the effectiveness of ComMU on stage1 of combinatorial music generation. During the inference phase, we insert chord tokens in order to infuse chord progression (Figure 5).

### 4.2 Evaluation metric

It is worth mentioning that the objective evaluation of generated music quality is still an open issue [35]. However, the goal of our task, which is to generate various high-quality music while being well controlled by metadata, is clear. In line with these goals, we evaluate the generated music on three criteria: controllability, diversity, and fidelity. We generate samples one by one from the validation metadata to measure the controllability. For measuring diversity, we generate 10 samples per one validation metadata.

**Controllability.** We evaluate how accurately the given metadata controls the generated music. We focus on pitch, velocity, and harmony(key, chord), which are clearly measurable among the 12 metadata. The controllability of the pitch is measured by the ratio of notes that meets the given pitch range among all generated note sequences. The number of notes within the min/max velocity range

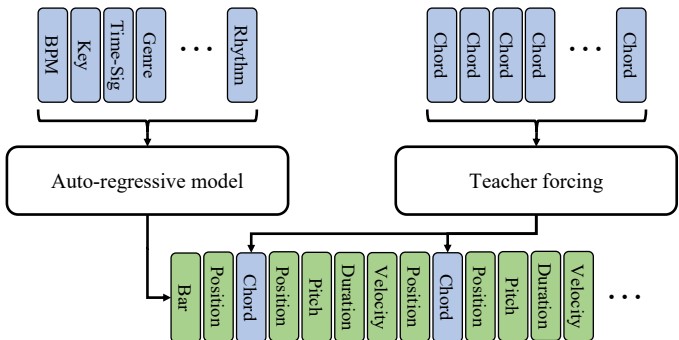

Figure 5: Architecture for the `stage1` of combinatorial music generation at inference phase.

| K, $\tau$ | Controllability | | | Diversity |
| | CP ↑ | CV ↑ | CH ↑ | D ↑ |
| --- | --- | --- | --- | --- |
| 32, 0.7 | 0.8798 | 0.9696 | 0.9976 | 0.2626 |
| 32, 0.95 | 0.8412 | 0.9102 | 0.9946 | 0.3160 |
| 32, 1.2 | 0.7721 | 0.8566 | 0.9910 | 0.3688 |
| 100, 0.95 | 0.8609 | 0.9138 | 0.9954 | 0.3195 |
| 200, 0.95 | 0.8488 | 0.9043 | 0.9958 | 0.3165 |

(a) Objective metric result

(b) Win rates of generated vs. ground-truth(%)

Table 3: Evaluation results. Objective metric include pitch control (CP), velocity control (CV), harmony control (CH), and diversity (D). We measure win rates using generated samples with K: 32, $\tau$: 0.95.

is used to measure the controllability of velocity. For evaluating harmonies, we check whether the pitch of note is within the scale of the corresponding audio key. If the pitch is out of scale, we check whether the pitch matches the chord tone over the duration. If both conditions are not satisfied, the note is evaluated as dissonant. The number of notes evaluated as not dissonant is used as the metric to quantify the controllability of the harmony.

**Diversity.** We define diversity metric as the average pairwise distance between multiple music generated from the same metadata. Although there is no general way to measure the distance between music, it can be defined by using chroma [36] and groove similarities [37], which measure the cosine similarity of pitch class and rhythm between two music [18]. We define the distance and the diversity metric as follows:

$$\texttt{dist}(\texttt{o}_\texttt{i}, \texttt{o}_\texttt{j}) = \sqrt{\frac{(1 - \texttt{sim}_{\text{chr}}(o_i, o_j))^2 + (1 - \texttt{sim}_{\text{grv}}(o_i, o_j))^2}{2}}, \quad (3)$$

$$\texttt{diversity}(\mathcal{O}) = \frac{1}{\binom{n}{2}} \sum_{i=1}^{n} \sum_{j=1}^{n} \texttt{dist}(o_i, o_j) \ , \ i < j, \quad (4)$$

where $\mathcal{O} = \{o_i\}_{i=1}^{n}$, $\texttt{sim}_{\text{chr}}$, and $\texttt{sim}_{\text{grv}}$ denote respectively $n$ piece of generated music with same metadata, chroma similarity and groove similarity.

**Fidelity.** We define the fidelity as win rates of generated samples against real samples. To this end, we have conducted a survey where participants were asked to select their preference between the generated and the real sample with the same metadata.

### 4.3 Results

In the language model, the sampling parameter top-k ($K$) and the softmax temperature ($\tau$) [38] affect the generated sequences. We explore the controllability and the diversity of generated music by adjusting $K$ and $\tau$. Table 3(a) shows that the controllability declines and the diversity rises as $\tau$ increases. However, the parameter change does not affect the controllability of the harmony because

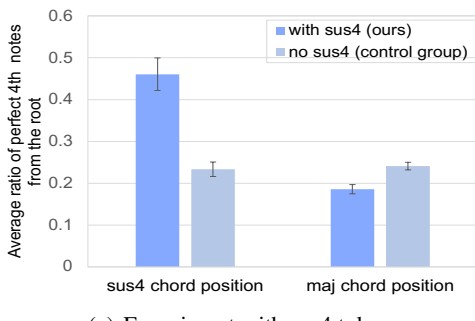 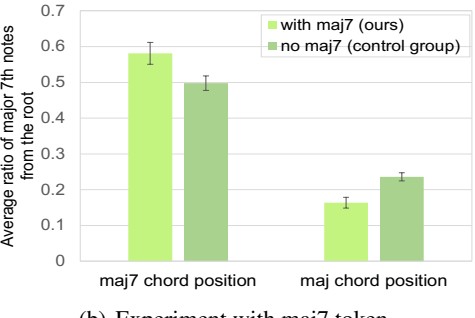

(a) Experiment with sus4 token      (b) Experiment with maj7 token

Figure 6: **Ablation study results of chord quality.** We take sus4 and maj7 chord quality for all chord roots, respectively. Sus4 is a chord quality in which the major 3rd is replaced with a perfect 4th. Maj7 is consisted of the root, major 3rd, perfect 5th, and major 7th. Therefore, compared to the major chord (root, major 3rd, perfect 5th), the sus4 can produce perfect 4th notes, and the maj7 can produce major 7th notes.

the explicit chord progression induces accurate musical coherence. Moreover, unlike text generation, even if K increases, the change in the results is negligible. This is because the vocabulary size of music (i.e., ComMU) is way smaller than a general language model. Empirical results reveal discrepancies between language and music, and we hope that research on model architecture suitable for music generation will be conducted in the future.

For evaluating fidelity, we ask 40 anonymous composers to 30 questions comparing generated samples to ground-truth samples with the same metadata through Amazon Mechanical Turk. Table 3(b) shows the average of win rates of generated samples against real samples. This shows how close the generated music is to the human level. It does not beat the ground-truth samples yet, but it shows a fairly close performance. See Appendix I for details about Amazon Mechanical Turk. Qualitative results can be found in `https://pozalabs.github.io/ComMU/`.

## 5 Discussion

Our rich metadata makes it easy to reflect the composer's intention and increases the capacity and the flexibility of the automatic composition. Since track-role and extended chord quality are primary metadata that distinguishes ComMU from other datasets, we demonstrate the advantage of unique metadata through ablation studies.

### 5.1 Extended chord quality

Chord quality of ComMU not only has major(maj), minor(min), diminished, augmented, and dominant but also incorporates sus4, maj7, half-diminished, and min7. Extended chord quality makes it easier to understand the precise intention of composition and leads to the improvement in harmonic performance. We experiment with how the extended chord quality affects the note creation.

In this experiment, we compare the note sequence produced by two different models where the experimental group (ours) holds sus4 or maj7 tokens while the control group does not. The chord ground truth depends on whether the composer initially intended to write a sus4 or maj7 chord in a particular position in the sequence. Here we compare the average ratio of perfect 4th and major 7th notes present in those positions depending on their chord ground truth. In other words, the right halves of both Figure 6(a) and 6(b) refer to the chord ground truth of the major chord, whereas the left halves of both 6(a) and 6(b) refer to the chord ground truth of sus4 and maj7 chord respectively.

As shown in Figure 6(a), the experimental group better adheres to the sus4 chord scale, where the average ratio of perfect 4th notes produced is more than 0.2 higher in the experimental than in the control group. In the case of major chord position, the experimental group produced less perfect 4th notes than the control group, better conforming to the major chord scale than its counterpart do. Such is also true in the case of major7 chords in Figure 6(b). This demonstrates that whether the composer

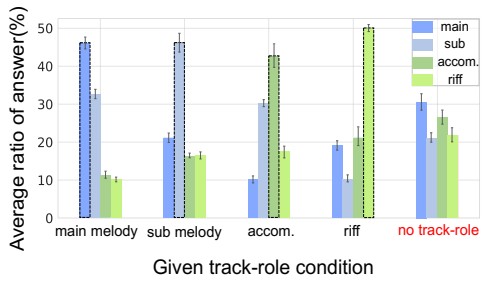

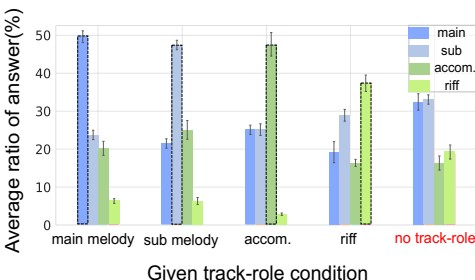

(a) Response distribution of piano samples          (b) Response distribution of string samples

Figure 7: Response of track-role classification survey. It illustrates that appropriate music is created when the role of each track is specified as a condition.

intended to write major or extended chords (sus4 and maj7) in a particular position in the sequence, the model trained with the extended chord tokens better reflects his or her intention.

## 5.2 Multi-track with track-role

While early literature has configured multi-track with instruments [19, 20], ComMU introduces track-role to configure multi-track. This can improve the capacity and flexibility of automatic composition and give more detailed conditions for music generation. For example, we can make music consisting of piano and guitar without track-role information. However, we cannot make music with piano as an accompaniment and guitar as a melody track. This is because even the same instrument has a different note shape, pitch range, and rhythm depending on the track-role(Figure 4).

To demonstrate the impact of the track-role, we conduct experiments comparing generated music, including track-role, with music that does not. We play 64 generated music samples through Amazon Mechanical Turk to 20 anonymous professional composers worldwide and ask which track-role is most appropriate for each music.

According to the survey result in Figure 7, if no track-role information is provided, the response resembles the track-role distribution by the instrument depicted in Figure 4. This indicates that the track-role is generated at random. Conversely, when a track-role is given, many subjects respond that the given track-role is the most appropriate. This indicates that our track-role metadata provides significant guidance to the generated music.

## 6 Conclusion

In this paper, we attempted to push the boundaries of automatic composition by introducing combinatorial music generation. We presented ComMU, a dataset for combinatorial music generation `stage1`, consisting of 12 metadata matched with note sequences manually created by composers. Furthermore, we present quantitative evaluations for generated music through fidelity, controllability, and diversity, and we demonstrate the benefits of unique metadata such as track-role and extended chord quality. We still need the generated note sequences to be combined by experts in `stage2`, but by automating `stage1`, we have dramatically reduced the time it takes to compose music. With the development of `stage2` in the future, we expect the automatic composition to be close to the human level. It is worthwhile to notice that combinatorial music generation would be one of the potential uses of the ComMU dataset. We hope that ComMU opens up a wide range of future research on automatic composition.

**Acknowledgment.** This research was supported by Culture, Sports and Tourism R&D Program through the Korea Creative Content Agency grant funded by the Ministry of Culture, Sports and Tourism in 2022 (Project Name: AI Producer: Developing technology of custom music composition, Project Number: R2022020066, Contribution Rate: 100%).

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
