# OpenReview forum: "ComMU: Dataset for Combinatorial Music Generation"
_NeurIPS.cc/2022/Track/Datasets_and_Benchmarks — NeurIPS 2022 Datasets and Benchmarks _

### Official Review · Reviewer_E3EV · 2022-07-06
**A unique combinatorial music generation dataset informed by expert opinions**

**Rating:** 7
**Confidence:** 4
**Correctness:** seems ok
**Clarity:** Paper is very well organized and pres…

**Strengths:**

1. The dataset is constructed by expert opinions, thus making it more reliable and robust.

2. Offers rich meta data including extended chord quality and track role, taking into account the intentions of the composers.



**Weaknesses:**

1. Music genres and instruments explored are largely dominated by Western cultures.

2. Paper can benefit from providing a detailed data sheet - please see comments below.

**Additional Feedback:**

Please see responses above.

**Documentation:**

Sufficient details are provided. That said, the authors might want to consider providing a data sheet and/or an art sheet for the dataset- please refer to the papers titled "Datasheets for Datasets" and "Artsheets for Art Datasets".

**Ethics:**

The authors acknowledge some of the ethical concerns that can arise in such applications --marginalization of music cultures ( especially given that most studies focus on Western instruments and music genres)  apart from issues related to copyrights and licensing. Thus, it becomes useful to provide an art sheet/datasheet for the paper including questions related to data generation, use, distribution, composition, provenance and so on.

**Relation To Prior Work:**

Although the work concerns music generation, it might be still useful to contextualize some related work in the space combinatorial music selection such as https://arxiv.org/abs/2104.12432. This paper may be a relevant prior work- https://arxiv.org/abs/2104.12432

**Summary And Contributions:**

The paper offers an excellent groundwork for furthering research in the space of human-computer co-creativity and more specifically in the context of combinatorial music generation. The proposed method values the intention of the composers, a factor that is crucial for the development of responsible art.  Furthermore, the dataset includes unique metadata that can augment the quality of generated music.

---

> ### Author Response · Authors · 2022-08-23
> **Response to Reviewer E3EV**
>
> We thank reviewer E3EV for the constructive comments and the time. We address the concerns and questions of the reviewer below:
>
> **Q1**. Music genres and instruments explored are largely dominated by Western cultures.
> - We are glad to see that you also thought of the bias of dataset toward Western cultures (Appendix G: Societal impact). We added a related sentence to appendix G in Lines 584-585. We have begun to collect oriental music (e.g., Korean traditional music). We will open it as soon as the data is available in the future.
>
> **Q2**. Paper can benefit from providing a detailed data sheet - please see comments below. it becomes useful to provide an art sheet/datasheet for the paper including questions related to data generation, use, distribution, composition, provenance and so on.
> - We already shared the datasheet in the supplementary material as “dataset_documentation.pdf”. I think it was difficult to recognize it because of the title, and we would appreciate it if you could check it.
>
> **Q3**. Although the work concerns music generation, it might be still useful to contextualize some related work in the space combinatorial music selection such as https://arxiv.org/abs/2104.12432. This paper may be a relevant prior work- https://arxiv.org/abs/2104.12432
> - The paper [https://arxiv.org/abs/2104.12432)] you mentioned was something we have missed, and it was nice to be able to think about the relevance with this research. We added it as a reference of our paper in Line 19, and we sincerely appreciate your recommendation.

---

### Official Review · Reviewer_VTd3 · 2022-07-13
**Valuable dataset for Interesting combinatorial music generation tasks**

**Rating:** 7
**Confidence:** 3
**Correctness:** Yes
**Clarity:** This paper is well written.

**Strengths:**


(1)  The dataset is manually made by professional composers;

(2)  The dataset includes two unique metadata, track and extended chord quality, that can improve the capacity of the automatic composition.

(3) The dataset includes abundant metadata (12 metadata), while the previous works generate music with much fewer metadata.

**Weaknesses:**

In the experiment, the authors only use one baseline (Transformer-XL). Therefore, all conclusions are drawn from this baseline. It's important add some baselines to enhance the conclusions.

**Additional Feedback:**

I'm not familiar with the field of automatic music composition. Here are some questions I'm curious about.

(1) What are the difficulties and challenges of compositional music generation tasks?

(2) How this article can help researchers in the field?

(3) What is the relationship between combinatorial music generation and automatic music composition?

Here are some further questions about the experiment.

(1) In Figure 7, there is no error bar or confidence intervals.

(2) In Table 3, the authors keep the parameters ($K$ and $\tau$) fixed. This makes me a little confused. If the parameters are fixed, what is the role of the validation set? Are there other parameters that need to be selected through the validation set?

(3) Why make all the metadata in the validation set exclusive to the training set?

(4) Why are the evaluation metrics in Figures 6 and 7 different from those in Table 3?

**Documentation:**

Yes

**Relation To Prior Work:**

Yes

**Summary And Contributions:**

This paper introduces an interesting new task called combinatorial music generation, and provides a symbolic music dataset ComMu for illustration. The ComMU dataset is manually constructed by professional composers and includes 12 musical metadata, two of which, track and extended chord quality, are unique metadata in this dataset.

---

> ### Author Response · Authors · 2022-08-23
> **Response to Reviewer VTd3**
>
> We thank reviewer VTd3 for the review and the suggestion. We address the concerns and the questions of the reviewer below:
>
> **Q1**. In the experiment, the authors only use one baseline (Transformer-XL). Therefore, all conclusions are drawn from this baseline. It's important add some baselines to enhance the conclusions.
> - There are experimental results and analyses in the additional baseline model (transformer-GAN) in appendix E.2. Our paper focuses on presenting the method for solving combinatorial music generation and the standard for symbolic music datasets. It would be great to compare the performance between different models in future work.
>
> **Q2**. What are the difficulties and challenges of compositional music generation tasks?
> - The most difficult challenge would be to create various and high-quality music while accurately reflecting the composer’s intention through metadata. We presented a new task (combinatorial music generation) and a dataset to address this problem.
>
> **Q3**. How this article can help researchers in the field?
> - First of all, in the combinatorial music generation task we presented, diverse and high-quality music can be generated and be accurately controlled by rich metadata provided in the ComMU dataset. Also, there are two important and distinctive properties of our dataset: the introduction of suspended chord quality and track-role information. Before ComMU, there has been no dataset that presents the two while combining all 12 metadata at once. In other words, we opened a new field of research on generating diverse music with a high level of controllability.
>
> **Q4**. What is the relationship between combinatorial music generation and automatic music composition?
> - Combinatorial music generation is a task to solve conditional music generation, a type of automatic music composition, receiving certain musical metadata as input. Our task has the advantage of being able to produce a variety of clearly controlled music through rich metadata.
>
> **Q5**. In Figure 7, there is no error bar or confidence intervals.
> - We add confidence intervals in figure 7. We appreciate your recommendation.
>
> **Q6**. In Table 3, the authors keep the parameters ($K$ and $τ$) fixed. This makes me a little confused. If the parameters are fixed, what is the role of the validation set? Are there other parameters that need to be selected through the validation set?
> - $K$ and $τ$ are fixed only for the win-rate experiment in subjective evaluation, which compares generated music with ground truth, and not for evaluating the objective metric. We selected values of $K$ and $τ$ that performed most generally well in the object metric and used as fixed $K$ and $τ$ in the win-rate experiment to ensure the most general subjective evaluation possible.
>
> **Q7**. Why make all the metadata in the validation set exclusive to the training set?
> - We believe that it is a more general and robust way to evaluate whether our system works well with metadata that model has never seen before. We wanted to show that the model can learn certain rules for composition and print notes even with new metadata.
>
> **Q8**. Why are the evaluation metrics in Figures 6 and 7 different from those in Table 3?
> - It is because the evaluation targets are different. Table 3(a) is for the controllability metric to measure how well each metadata value was preserved and the diversity metric to see how various the generated music are. On the other hand, Figures 6 and 7 are to evaluate the benefit of suspended chord quality and track-role, respectively. Therefore, both Figure 6 and 7 used different metric since they cannot be evaluated by objective metric (controllability, diversity in Table 3).

---

### Official Review · Reviewer_7zFc · 2022-07-25
**High-quality music dataset with rich metadata**

**Rating:** 7
**Confidence:** 3
**Correctness:** The paper seems correct and claims we…
**Clarity:** The paper is well-written and easy to…

**Strengths:**

1. The music dataset, collected by professional composers, is of such high-quality that it can be used commercially.
2. It contains rich and unique metadata which is helpful controllable and diverse music generation.
3. Documentation and materials including website and GitHub repository are well-organized.


**Weaknesses:**

1. The dataset lacks a big of genre diversity since there are only two genres (cinematic and new-age). More diversity in genre will strengthen the usefulness.
2. The paper mainly focuses on stage 1 so it lacks description on stage 2. I think more contents of stage 2 would strengthen the completeness of this work, e.g., Are there any patterns or tendencies of professional composers? How about almost randomly generated combination compared to manually generated one?

**Additional Feedback:**

1. (line 86) The reference for ADL Piano MIDI is missing, while one is correct in Table 1.
2. (Table 1) LakhMIDI -> Lakh MIDI
3. (line 105) Where do "reference music" come from? Do they utilize external music sources such as online free music repository?
4. (Eq 1) I think $\sum^T_{t=12}\log p_\theta(x^S_t|x_{\lt t})$ is more accurate, because metadata consists of $x_{1:11}$.

**Documentation:**

The documentation contains abundant and detailed information.

**Ethics:**

No.

**Relation To Prior Work:**

The relation to prior work and contribution are clearly discussed.

**Summary And Contributions:**

This paper introduces MIDI-based symbolic music dataset, ComMU, with rich and unique metadata for combinatorial music generation which consists of two stages; 1) note sequence generation given metadata, and 2) combination of the sequences for one completed music. The authors quantitatively analyze the dataset and evaluate Transformer baseline in diversity, controllability, and fidelity. The dataset seems useful for music generation field and combinatorial music generation task seems promising approach for high-quality music generation.

---

> ### Author Response · Authors · 2022-08-23
> **Response to Reviewer 7zFc**
>
> We thank the reviewer 7zFc for your comments and time. We discuss the concerns and questions from the reviewer below.
>
> **Q1**. The dataset lacks a big of genre diversity since there are only two genres (cinematic and new-age). More diversity in genre will strengthen the usefulness.
> - We agree with your point. As stated in Appendix C, metadata such as time signature and rhythm is also imbalanced. If they are collected more diversely, the value of the dataset will be maximized. It would be great if you consider our study as the starting point, providing a standard for music datasets. We have collected additional data from various genres such as jazz and Korean traditional music and plan to release it later. We would greatly appreciate if you continue to follow our research.
>
> **Q2**. The paper mainly focuses on stage 1 so it lacks description on stage 2. I think more contents of stage 2 would strengthen the completeness of this work, e.g., Are there any patterns or tendencies of professional composers? How about almost randomly generated combination compared to manually generated one?
> - As mentioned in the conclusion, we are considering stage2 as a future work. However, we have compared randomly generated combination to manually generated one as it seemed very interesting. We added the results of the comparison on our demo page (https://pozalabs.github.io/ComMU/).
> - We briefly explain how the composers combine samples in stage2. Sample combination methods are diverse and extensive. In the case of the example song in demo page, start based on the accompaniment track and layer the main melody track on it. For the diversity of the song, there are ways to add sub-melody or change the instrument used by the main melody, and add pad track or riff track to give the dynamic of the song.
>
> **Q3**. (line 86) The reference for ADL Piano MIDI is missing, while one is correct in Table 1.
> - We correct typos. Thanks.
>
> **Q4**. (Table 1) LakhMIDI -> Lakh MIDI
> - We correct typos. Thanks.
>
> **Q5**. Where do "reference music" come from? Do they utilize external music sources such as online free music repository?
> - Any music can be reference music. Generally, a client gives some reference music and asks a composer to compose a similar kind of music. Then the composer creates new sequences with a similar atmosphere. We tried to show that we could do this with only 12 metadata.
>
> **Q6**. (Eq 1) I think $\sum_{t=12}^{T}\log ⁡p_{θ}(x_t^S|x_{<t})$ is more accurate, because metadata consists of $x_{1:11}$.
> - We accept your suggestion and change our equation 1 in page 6 to clarify our paper. Thanks.

---

> > ### Comment · Reviewer_7zFc · 2022-08-25
> > **Response to Rebuttal**
> >
> > Thanks for addressing the most of concerns. I was surprised that musics from random combination are such a high quality as well. But I still have a concern for the "reference music". If the reference music can be any music, isn't there any possibility of plagiarism issue?

---

> > > ### Author Response · Authors · 2022-08-25
> > > **Response to Reviewer 7zFc**
> > >
> > > The plagiarism issue is one of the things we care about the most. We divided composers into analysis and sample-making groups to fundamentally prevent plagiarism issues. The former creates objective guidelines for composition from reference music and delivers them to the latter group. The latter group does not see the note sequence of reference music and composes only with objective guidelines derived from it, so it creates a sample independent of reference music. In addition, the former group double-checks for plagiarism issues in the samples made by the latter group.
> > >
> > > Thanks for your careful reviews.

---

> > > > ### Comment · Reviewer_7zFc · 2022-08-26
> > > > **Thanks for your response**
> > > >
> > > > Now all of my concerns are resolved. Thank you.

---

### Official Review · Reviewer_y1Pe · 2022-07-27
**Professionally created 11,144 MIDI samples annotated with 12-dimensional meta data**

**Rating:** 7
**Confidence:** 3
**Correctness:** There are no issues with correctness.
**Clarity:** The paper is very clearly written.

**Strengths:**

1. The paper provides O(10^4) samples of short musical compositions with 12-dimensional annotations to the community.
2. There is an interesting baseline solution presented for the problem of combinatorial music generation using this data set.
3. All the code, data set and website are of high quality.

**Weaknesses:**

1. I am not an expert in music. So, I cannot comment on the academic significance of the problem in music. However, as an end user, I am quite happy with the quality of music being generated here.

**Additional Feedback:**

This is wonderful work put together in a complete and clear manner. I greatly enjoyed reading the paper and browsing the related website.

**Documentation:**

The benchmark code and the data set are both well documented.

**Ethics:**

There are no significant concerns about ethics.

**Relation To Prior Work:**

The paper cites related work in the area.

**Summary And Contributions:**

The paper discussed a new data set of 11.144 short music samples created by professional human experts and annotated with 12-dimensional metadata. The metadata is then used to investigate a combinatorial analysis and synthesis of longer music sequences. This is a very interesting and new line of work.

---

> ### Author Response · Authors · 2022-08-23
> **Response to Reviewer y1pe**
>
> We are grateful for your review of our paper as an end user, confirming that our research can be used practically for the crowd as well as for those in the academia. We add random combination of music in stage2 to our demo web page to show that end users can create and be inspired by generating music even if they are non-experts. Hope you enjoy it (https://pozalabs.github.io/ComMU/).

---

> > ### Comment · Reviewer_y1Pe · 2022-09-04
> > **Thank you for your feedback.**
> >
> > The demo page is indeed useful in understanding and fully appreciating the impact of the work effortlessly.

---

### Official Review · Reviewer_maYJ · 2022-07-27
**Review for ComMU: Dataset for Combinatorial Music Generation**

**Rating:** 7
**Confidence:** 4

**Strengths:**

This paper provides a valuable MIDI dataset for combinatorial music generation. Music generation has recently attracted much attention in the community since the success of generative models. However, the publicly avaible music data for research on this topic are lacking due to the copyright and license issues. While previous research has open some datasets for generative music continuation, little has been done for  controllable music generation. To bridge this gap, this paper provides a valuable MIDI dataset with abundant meta features. Meanwhile, the paper demonstrates the use of the dataset with combinatorial music generation tasks. The dataset and generated samples are availble. I checked the generated samples on the website, the quality is high which verfies the usefulness of the dataset. The dataset should benefit the community to perform research on controllable music generation.





**Weaknesses:**

The paper is well written and easy to follow. My concern is that in Section 4.1 the authors define the task as a autoregressive generation problem. But it does not clarify its relationship to existing work and why not implement an exsiting approach as the baseline.

**Additional Feedback:**

I suggest to specify a clear license for the code on github.

**Clarity:**

The paper is overall well-written. But it would be better to provide more background information for readers who have no prior knowledge in the field.

**Correctness:**

This paper provides a dataset. The dataset contains 12,025 MIDI files. The only question is how the number of samples is determined and whether they are sufficient for commercial-level music generatoin study?

**Documentation:**

After checking the website and github pages, the documentation looks sound for me.

**Ethics:**

The datasets are MIDI files, which does not contain voice information. The license is also explictly specified. It should have no ethics issues.

**Relation To Prior Work:**

The paper present a new set of MIDI files with detailed meta data. The dataset is new to the research on controllable music generatoin task. Also, the author compare the dataset with previous ones in Table 1. For the methodology part, the authors define the task as a autoregressive generation problem. Yet, its relationship to existing work is lacking.

**Summary And Contributions:**

This paper provides a dataset for combinatorial music generation. The dataset consists of 12,025 MIDI files with 12 musical metadata (bpm, genre, key, instrument, track-role, time signature, 48 pitch range, number of measures, chord progression, min/max velocity, and rhythm). The paper takes combinatorial music generation as a two-phase problem: note sequence generation and multiple note sequences combination. The dataset is aimed for the first phase, note sequence generation with metadata. The authors formulate it as a conditional generation task which generate note sequence autoregressively conditioned on the metadata provided. Some data analyses and baseline eperiments have been conducted to validate the usefulness of the provided dataset.

---

> ### Author Response · Authors · 2022-08-23
> **Response to Reviewer maYJ**
>
> We thank reviewer maYJ for the review and the suggestion. We address the concerns and the questions from the reviewer below:
>
> **Q1**. My concern is that in Section 4.1 the authors define the task as a autoregressive generation problem. But it does not clarify its relationship to existing work and why not implement an exsiting approach as the baseline
> > The paper present a new set of MIDI files with detailed meta data. The dataset is new to the research on controllable music generatoin task. Also, the author compare the dataset with previous ones in Table 1. For the methodology part, the authors define the task as a autoregressive generation problem. Yet, its relationship to existing work is lacking.
>
> - Existing studies done to solve music generation problem can largely be divided into transformer-based autoregressive models [1, 2, 3, 4], GAN [5, 6] and VAE [7, 8] models. There are also studies such as Transformer-GAN [9] and MuseMorphose [10] that combine them. The task we presented is a conditional music generation problem that generates music corresponding to given metadata. The most prevalent model used to solve the conditional music generation problem is an autoregressive model based on the transformer [11], hence the baseline. The difference between the proposed problem in 4.1 and the existing autoregressive approach is that a teacher-forcing structure for feeding in chord progression as a condition has been added to the interference stage.
>
> - We wrote the results of additional experiments with Transformer-GAN in appendix E.2 for comparison with the existing approach. While our work proposes method and dataset standard in tackling combinatorial music generation, performance comparison and analysis between various models may well be addressed in future work.
>
> **Q2**. The only question is how the number of samples is determined and whether they are sufficient for commercial-level music generation study?
> - Results in table 3(b) and 4.3 indicate that the generated music shows comparable quality to the actual sample. This shows that while it may not be perfect, the number of samples in our dataset is sufficient for commercial-level music generation study. We expect to generate commercial-level music even closer to the human level when a more diverse data is collected, based on the standard proposed in the paper.
>
> **Q3**. But it would be better to provide more background information for readers who have no prior knowledge in the field.
>
> - For those who do not have background knowledge in the music field, a visual explanation of midi data and ComMU data representation is provided in the supplementary video. In addition, we reinforced the metadata description in Appendix B and created a new section to explain music terminologies in Appendix J. Thank you for your feedback. We were able to polish the paper so that people without background knowledge could also understand.
>
> **Q4**. I suggest to specify a clear license for the code on GitHub.
> - It's what we were missing out. Thanks to you, we add the license on code on GitHub.
>
> ***
> [1] Huang et al., Music Transformer: Generating Music with Long-Term Structure, ICLR 2019.
>
> [2] Payne et al. Musenet. OpenAI Blog, 3, 2019.
>
> [3] Huang et al., Pop Music Transformer: Beat-based Modeling and Generation of Expressive Pop Piano Compositions, ACM International Conference on Multimedia 2020.
>
> [4] Hsiao et al., Compound Word Transformer: Learning to Compose Full-Song Music over Dynamic Directed Hypergraphs, AAAI 2021.
>
> [5] Dong et al., MuseGAN: Multi-track Sequential Generative Adversarial Networks for Symbolic Music Generation and Accompaniment, AAAI 2018.
>
> [6] Dong et al., Convolutional Generative Adversarial Networks with Binary Neurons for Polyphonic Music Generation, ISMIR 2018.
>
> [7] Roberts et al., A Hierarchical Latent Vector Model for Learning Long-Term Structure in Music, ICML 2018.
>
> [8] Brunner et al., MIDI-VAE: Modeling Dynamics and Instrumentation of Music with Applications to Style Transfer, arXiv preprint 2018.
>
> [9] Muhamed, Aashiq et al., Symbolic music generation with transformer-gans, AAAI 2021.
>
> [10] Wu et al, MuseMorphose: Full-Song and Fine-Grained Music Style Transfer with One Transformer VAE, arXiv preprint 2021.
>
> [11] Rutte et al., FIGARO: Generating Symbolic Music with Fine-Grained Artistic Control, arXiv preprint 2022.

---

### Official Review · Reviewer_2G3b · 2022-07-27
**Dataset for combinatorially creating music samples based on given conditions**

**Rating:** 7
**Confidence:** 4

**Strengths:**

- The paper is a good contribution to datasets for machine learning research in music, especially with music generation. This is a worthwhile contribution given that music datasets have not advanced as rapidly as other areas.
- The proposed dataset contains several metadata that provide a decent amount of interpretiveness to the music and control over generated music using combinatorial music generation.

**Weaknesses:**

- One of the claims of the proposed dataset is the diversity of the music samples, but that is difficult to justify given that the samples were manually generated by a constrained set of composers. Plus, no information is given about the composers, so the compositions may be limited by the composers' skillset or orientation (e.g., genre preference, chord diversity, etc.)
- There is a lot more to music than just chord progressions and the other metadata contained in the proposed approach. As such, the generality might be quite limited, e.g., in the absence of other musical qualities like scales, arpeggios, etc.
- Like the prior point, chord progressions are extremely complex and diverse. It would have been nice to get a sense of the kinds of progressions that were included in the dataset and what the patterns represent or how the patterns were created.
- An apparent major contribution of the paper is combinatorial music generation, but very little time is spent on describing it in the paper. It seems to me like the main contribution is the dataset and combinatorial music generation would be one of the potential uses of the dataset. Unless I'm mistaken, the paper may need to be restructured to better represent the contributions (some minor restructuring of sentences in the abstract/intro may solve this issue).

**Additional Feedback:**

- How do you ascertain the quality of the music samples, given the apparent anonymity of the composers?
- It seems like the quality of the samples would be very subjective. For example, genres are very subject to interpretation by the composers (e.g., one composer may describe new age music as cinematic). Or, a riff would vary substantially based on the skill of the composer. This could have implications for training, wherein a desire for cinematic music may be misinterpreted due to a different interpretation in the training data. How would this be accounted for in the dataset?

**Clarity:**

The paper is generally well-written and easy to understand. There are some typos, like an undefined reference on line 86. There is a disparity in the number of MIDI samples between line 46 and 97.

**Correctness:**

The claims made in the paper seem correct. Some clarification might be helpful for any rules that were followed in the composition of the music samples. Or, was it arbitrary?

**Documentation:**

Sufficient information is provided on the data collection and organization, availability, and maintenance, and ethical and responsible use.

**Ethics:**

No noted ethical concerns.

**Relation To Prior Work:**

The paper clearly distinguishes the proposed work from prior work. The main difference is that prior symbolic music datasets do not have enough metadata for sophisticated control.

**Summary And Contributions:**

The paper proposes combinational music generation, a task for creating different background music based on given conditions. The task creates music samples with musical metadata and combines them to produce complete music. The paper also introduces 'ComMU', a symbolic music dataset consisting of 12,025 short music samples and their metadata. The dataset is manually created by professional music composers and features 12 musical metadata that embrace the composers' intentions.

---

> ### Author Response · Authors · 2022-08-23
> **Response to Reviewer 2G3b [2]**
>
> **Q5**. The claims made in the paper seem correct. Some clarification might be helpful for any rules that were followed in the composition of the music samples. Or, was it arbitrary?
> - We provided the process and standards of making music samples in chapter 3.1 of main paper and chapter 3 of the datasheet (supplementary material).
>
> **Q6**. There are some typos, like an undefined reference on line 86. There is a disparity in the number of MIDI samples between line 46 and 97.
> - We corrected typos. Thank you.
>
> **Q7**. How do you ascertain the quality of the music samples, given the apparent anonymity of the composers?
> - Professional composers with three or more years of experience wrote the samples and have double-checked them to ensure the samples met the objective standards. We are unable to disclose personal information about the composers as of now, but we hope to in the future on the project page after we get the consent.
>
> **Q8**. It seems like the quality of the samples would be very subjective. For example, genres are very subject to interpretation by the composers (e.g., one composer may describe new age music as cinematic). Or, a riff would vary substantially based on the skill of the composer. This could have implications for training, wherein a desire for cinematic music may be misinterpreted due to a different interpretation in the training data. How would this be accounted for in the dataset?
> - Objective criteria were created for each metadata to prevent arbitrary interpretation by composers. For example, we define the new age as a melodious genre mainly consisting of keyboard instruments and small-scale instruments such as acoustic instruments. On the other hand, we define the cinematic genre is a large-scale genre with orchestra, especially involving classical instruments such as string ensembles in charge of the melody and the accompaniment (See 3.3 Metadata in paper). We also made sure that if one composer composed music samples, other composers would check if the composition met the objective standard.

---

> ### Author Response · Authors · 2022-08-23
> **Response to Reviewer 2G3b [1]**
>
> We thank reviewer 2G3b for the review and the suggestion. We address the concerns and the questions from the reviewer below:
>
> **Q1**. One of the claims of the proposed dataset is the diversity of the music samples, but that is difficult to justify given that the samples were manually generated by a constrained set of composers. Plus, no information is given about the composers, so the compositions may be limited by the composers' skillset or orientation (e.g., genre preference, chord diversity, etc.)
> - Recognizing the importance of the mentioned issue of diversity and the composers musical orientation, we have constructed the dataset strictly following the objective composition guidelines, not the composer’s taste. In addition, only samples that meet the criteria when double-checked by composers are included in the dataset. We add the explanation to supplementary material (see the datasheet in Lines 82-84) for further clarification. Thanks.
> - The diversity of our music samples induces the system the ability to generate various music even with the same metadata. Check the diversity of generated music samples on our demo page (https://pozalabs.github.io/ComMU/).
> - Professional composers with three or more years of experience wrote the samples and have double-checked them to ensure the samples met the objective standards. We are unable to disclose personal information about the composers as of now, but we hope to in the future on the project page after we get the consent.
>
> **Q2**. There is a lot more to music than just chord progressions and the other metadata contained in the proposed approach. As such, the generality might be quite limited, e.g., in the absence of other musical qualities like scales, arpeggios, etc.
> - Increasing the number of metadata for the sake of diversity may result in the sacrifice of controllability. However, adding other musical qualities such as scale and arpeggio as metadata and collecting more samples may provide the ability to reflect a more sophisticated compositional intentions develop a general music generation system.
> - Currently, we are in our course to secure a more diverse metadata and are willing to open it in future work when available. Our study is meaningful in that it presents a new standard for symbolic music dataset with metadata for music generation. We hope to see many follow-up studies, based on our paper, being conducted to build a richer music dataset.
>
> **Q3**. Like the prior point, chord progressions are extremely complex and diverse. It would have been nice to get a sense of the kinds of progressions that were included in the dataset and what the patterns represent or how the patterns were created.
> - Different chord progressions determine several elements of music. The pattern of chord progressions not only reflects the characteristics of a genre but also determines the mood of a music. The patterns in our dataset were learned and generated through commercial music. Im-Vm-VII-IV pattern is an example from our dataset.  This is a chord progression pattern taken from "Time" by Hans Zimmer from the movie "Inception" to create a tense and dark mood. This simple chord progression, suitable for mood expression, is widely used in the cinematic genre.
>
> **Q4**. It seems to me like the main contribution is the dataset and combinatorial music generation would be one of the potential uses of the dataset. Unless I'm mistaken, the paper may need to be restructured to better represent the contributions (some minor restructuring of sentences in the abstract/intro may solve this issue).
> - You understood our intentions correctly, and thus we add the sentence (combinatorial music generation would be one of the potential uses of the dataset) to the paragraph in Lines 293-295.

---

> > ### Comment · Reviewer_2G3b · 2022-08-23
> > **Thanks for your response**
> >
> > Thanks for your response. I was positive about the paper and remain so.
> >
> > To clarify, I don't think it's necessary to disclose the identity of the composers. Just some information about them, like their expertise, genre specialization, etc. For instance, a classical music composer would write very different music than a jazz music composer. The composer's expertise can have implications for the kinds of chord progressions that are generated.

---

### Author Response · Authors · 2022-08-23
**Revision Summary**

We appreciate the constructive reviews that help to make our paper stronger. We summarize the revision as follows:

- Clarification for rules that were followed in the composition of the music samples [2G3b] → Section 3 in the datasheet (supplementary material) on page 3
- Provide more background information for readers who have no prior knowledge in the music field [maYJ] → Appendix B on pages 13-14, Appendix J on pages 18-19
- How about randomly generated combination compared to manually generated one [7zFc] → Stage 2 in Demo website (https://pozalabs.github.io/ComMU/)
- Change Eq.1 for clarity [7zFc] → Section 4.1 on page 6
- Implement error bar in Figure 7 [VTd3] → We add error bar in Figure 7 on page 9
- Music genres and instruments explored are largely dominated by Western cultures [E3EV] → Appendix G on page 17
- Correct some typos [2G3b, 7zFc] → Thanks for your careful reviews.

We strongly believe that the suggestions and comments have made our submissions more compelling. We are more than glad to discuss anything about our paper, and will reflect it in the final version.

---

### Author Response · Authors · 2022-08-25
**Contributions & Strengths Summary**

Our contributions are (1) to provide ComMU, the first symbolic music dataset manually created by professional composers for automatic music composition with 12 rich metadata and (2) to propose the combinatorial music generation, the new task for the industry-level automatic music composition. We demonstrates the use of the dataset with combinatorial music generation tasks. Diverse and high-quality musics are created with our framework and dataset.

All reviewers mentioned about strengths of our paper:


- Our dataset is **manually made by professional composers** [VTd3, E3EV] and **valuable** to machine learning community [2G3b, maYJ] and **high quality** [y1Pe, 7zFc]  and provides **decent amount of interpretiveness** [2G3b] and contains **rich and unique metadata** [2G3b, maYJ, 7zFc, VTd3, E3EV].
- The paper, code and website are **very well written and organized** [2G3b, y1Pe, maYJ, 7zFc, VTd3, E3EV].

Recognizing these strengths is an absolute pleasure to us.

---

### Meta-Review · Area_Chair_zeWF · 2022-09-10

**Recommendation:** Accept
**Confidence:** 3

**Metareview:**

All six reviewers recommend acceptance. Reviewers found the large dataset of professional MIDI music and metadata to be a useful resource. Authors should attend to main points in the reviews when preparing a final version. These issues include those raised by Reviewer 2G3b around more clearly presenting the contributions, the description of stage 2 and equation typos (7zFc), and clarifying which baselines were run (VTd3). The AC sees no basis to overturn reviews, and thus recommends acceptance.

---

### Decision · Program_Chairs · 2022-09-16

Accept